# A Five-Week Periodized Carbohydrate Diet Does Not Improve Maximal Lactate Steady-State Exercise Capacity and Substrate Oxidation in Well-Trained Cyclists compared to a High-Carbohydrate Diet

**DOI:** 10.3390/nu16020318

**Published:** 2024-01-21

**Authors:** Gorka Prieto-Bellver, Javier Diaz-Lara, David J. Bishop, José Fernández-Sáez, Javier Abián-Vicén, Iñigo San-Millan, Jordan Santos-Concejero

**Affiliations:** 1Performance and Sport Rehabilitation Laboratory, Faculty of Sport Sciences, University of Castilla-La Mancha, 45071 Toledo, Spain; gorkapb.1@gmail.com (G.P.-B.); javier.abian@uclm.es (J.A.-V.); 2Institute for Health and Sport (IHeS), Victoria University, Footscray VIC 3011, Australia; david.bishop@vu.edu.au; 3Unitat de Suport a la Recerca Terres de l’Ebre, Fundació Institut Universitari per a la Recerca a l’Atenció Primària de Salut Jordi Gol i Gurina (IDIAPJGol), 43500 Tortosa, Spain; j.fernandez@idapjgol.info; 4Department of Human Physiology and Nutrition, University of Colorado, Colorado Springs, CO 80918, USA; isanmill@uccs.edu; 5Department of Physical Education and Sport, University of the Basque Country UPV/EHU, 48940 Leioa, Spain; jordan.santos@ehu.eus

**Keywords:** train low, carbohydrates, fat oxidation, performance, cycling, body composition

## Abstract

There is a growing interest in studies involving carbohydrate (CHO) manipulation and subsequent adaptations to endurance training. This study aimed to analyze whether a periodized carbohydrate feeding strategy based on a daily training session has any advantages compared to a high-carbohydrate diet in well-trained cyclists. Seventeen trained cyclists (*V*O_2peak_ = 70.8 ± 6.5 mL·kg^−1^·min^−1^) were divided into two groups, a periodized (PCHO) group and a high-carbohydrate (HCHO) group. Both groups performed the same training sessions for five weeks. In the PCHO group, 13 training sessions were performed with low carbohydrate availability. In the HCHO group, all sessions were completed following previous carbohydrate intake to ensure high pre-exercise glycogen levels. In both groups, there was an increase in the maximal lactate steady state (MLSS) (PCHO: 244.1 ± 29.9 W to 253.2 ± 28.4 W; *p* = 0.008; HCHO: 235.8 ± 21.4 W to 246.9 ± 16.7 W; *p* = 0.012) but not in the time to exhaustion at MLSS intensity. Both groups increased the percentage of muscle mass (PCHO: *p* = 0.021; HCHO: *p* = 0.042) and decreased the percent body fat (PCHO: *p* = 0.021; HCHO: *p* = 0.012). We found no differences in carbohydrate or lipid oxidation, heart rate, and post-exercise lactate concentration. Periodizing the CHO intake in well-trained cyclists during a 5-week intervention did not elicit superior results to an energy intake-matched high-carbohydrate diet in any of the measured outcomes.

## 1. Introduction

A nutritional strategy commonly used in endurance sports is to increase the consumption of carbohydrates [1], as low muscle glycogen availability can contribute to decreased exercise capacity [2]. Previous research has also associated high carbohydrate intake for athletes’ diets with improving specific parameters related to recovery, muscle damage, and performance [2,3,4]. In contrast to traditional guidelines [5], recent evidence suggests that the periodization of carbohydrates may benefit athletes due to the potential muscle cellular adaptations [6]. This strategy requires limiting exogenous carbohydrate intake before and/or during specific training sessions by a variety of techniques such as “sleep-low” (i.e., glycogen-depleting session of training is followed by overnight carbohydrate (CHO) restriction and a moderate-intensity session the subsequent morning), “fasted exercise” (i.e., performing endurance exercise in a fasted state, after an overnight fast and without CHO intake during session), or “twice-a-day training” (i.e., two sessions on the same day; the second is commenced with reduced muscle glycogen) [7].

Some studies found improvements in performance after the application of these nutritional protocols [8,9]. On the contrary, other studies do not report superior effects on performance [10,11,12,13,14,15,16]. The question arises whether a nutritional strategy in which carbohydrate intake is manipulated based on the goal, duration, and intensity of the training session, known as “fuel for the work required” [17], can be beneficial for endurance performance and whether it may produce an effect on substrate utilization in well-trained athletes during a 5-week period. Studies in which performance was measured following different carbohydrate (CHO) feeding strategies have usually tested recreational or trained athletes over 1–4 weeks of treatment [11,12,15]. Still, the study of prolonged carbohydrate periodization (5 weeks) in well-trained athletes (~70 VO_2peak_) is yet to be explored.

Similarly, most studies that applied CHO periodization performed a submaximal test followed by a maximal test to assess the participants’ performance [8,9,10,11,12,14,15]. The present study showed novelty in performing measurements at maximal lactate steady-state (MLSS) intensity. It is suggested to be the gold standard for assessing the lactate threshold, which is directly associated with physical capacity and performance, and accurately investigating the consequences of diet at the same physiological milestone [18,19].

Regarding substrate utilization and body composition, one of the most interesting and discussed outcomes when applied to CHO-restricted diets is increasing fat as fuel utilization to enhance endurance sports performance [11]. However, significant fat oxidation capacity improvements have been shown to be insufficient to observe improved performance in elite athletes compared with a high-CHO condition [13]. Further, low-carbohydrate diets are often used as weight-loss strategies by exercising individuals and athletes [8]. Several researchers reported a decrease in fat-free mass after low-carbohydrate diets for extended periods (>six months) [20]. However, when both interventions were performed over a short-term period (e.g., three weeks) with analogous daily calorie intake, markers of body composition were altered to a similar extent [21].

Therefore, this study aimed to investigate whether periodizing the carbohydrate intake during a 5-week period has any benefit compared to a traditional high-carbohydrate diet regarding performance, substrates used during exercise, and body composition in well-trained cyclists.

## 2. Materials and Methods

### 2.1. Participants

Seventeen male cyclists classified as highly trained [22] volunteered to participate in this study. All participants were training 15 to 20 h per week and competing in the U23 national-level cycling categories before the study. Participants undertook the study in early November after a two-week post-season break and were informed about the tests to be performed. All participants signed an informed consent form before participating in the study, which was approved by the local Research Ethics Committee (CEISH 113/2019).

All participants were randomly assigned to either a carbohydrate periodization (PCHO) group (*n* = 9) (age = 24.8 ± 8.3 years; peak oxygen uptake (*V*O_2peak_) = 71.9 ± 6.4 mL·kg^−1^·min^−1^; peak power output (PPO) = 379.1 ± 38.0 W) or a high-carbohydrate group (HCHO) (*n* = 8) (age = 28.2 ± 4.2 years; *V*O_2peak_ = 69.6 ± 6.8 mL·kg^−1^·min^−1^; PPO = 390.6 ± 21.9 W).

### 2.2. Study Design

This study used a parallel group design. All participants completed a pre–post-MLSS test intervention and performed a pre–post-intervention test to exhaustion (TTE). The PCHO and the HCHO groups completed the same endurance training program for five consecutive weeks (Table 1) but followed different feeding plans with similar energy intakes. The PCHO group performed 13 “low” sessions with low carbohydrate availability according to the low-intensity training sessions. The participants performed between two and three low-intensity sessions per week based on other studies which found significant changes in performance through this training protocol [8,17]. Furthermore, nutritional plans were prepared according to the exercise intensity schedule that was prescribed. Low-intensity sessions were performed at the training intensity described as Zone 1 (Z1)–Zone 2 (Z2), which was determined by the exercise intensity below the first lactate/ventilatory threshold, VTL1. Subjects did not have two low-intensity sessions consecutively (Table 1). In contrast, the HCHO group performed all training sessions with high-carbohydrate availability pre-, during and post-training. Both groups also completed nine strength training and thirteen moderate- to high-intensity bike training sessions during the intervention with high carbohydrate availability before, during, and after exercise. The nutritional structure of the feeding plans is described in Table 2. Low, medium, and high carbohydrate intakes were made for each meal depending on the training intensity as described elsewhere [17,23]. Selected whole-body measures and various strength, submaximal, and maximal performance tests were performed before and after the intervention. The study design is shown in Figure 1.

### 2.3. Preliminary Tests

Graded Exercise Test (GXT): The GXT was conducted to determine peak oxygen uptake (*V*O_2peak_) and peak power output (PPO). Each participant was tested on their bicycle attached to a Wahoo Kickr ergometer in the laboratory (Wahoo Fitness, model: WF113, Vietnam). *V*O_2peak_ was measured with a gas analyzer (Ergocard, Medisoft, Sorinnes, Belgium), and heart rate was measured with a heart rate monitor (Polar FT1, Helsinki, Finland). Participants completed an incremental protocol in which intensity was increased by 25 W every 2 min, starting at 100 W until exhaustion and using a freely chosen cadence [24]. The PPO (W) was calculated as PPO = W completed + 25 (t/120), where W is the last completed workload, and *t* is the number of seconds in the last workload [25].

Strength test: The mean propulsive velocity (MPV) was measured with a validated encoder (linear encoder, Chronojump Boscosystem, Barcelona, Spain) to determine individual force workloads during the intervention. Participants performed a half squat, with the test terminated when a load of 0.6 m·s^−1^ was reached, 80% of one-repetition maximum (RM) [26].

### 2.4. Pre–Post-Nutritional and Training Intervention Tests

Body composition: Body composition data were obtained in a fasted state pre- and post-intervention. All participants’ anthropometrical data were obtained using the ISAK methodology [27]. These included: a sum of 6 skinfolds circumferences and body mass (BM). Percentages of muscle mass and body fat were obtained by the Matiegka and Yuhasz formulas, respectively [28,29].

Maximal Lactate Steady State (MLSS): Participants completed at least three 30 min tests at a constant load in the laboratory pre- and post-intervention before the glycogen depletion test. The first MLSS test was carried out at 65% of the PPO and the following ones until the MLSS was found. Blood samples were obtained from the earlobe, and lactate concentrations were measured at the 10th and 30th minute (Lactate PLUS, Nova Biomedical GmbH, Mörfelden-Walldorf, Germany). Lactate values did not exceed 1 mmol·L^−1^ between the two measurements to find the power at which lactate remained steady [30]. The test was performed on the same ergometer as for the *V*O_2max_ test. Once the criteria for the MLSS intensity were met, a confirmation test was performed to verify that the intensity was correct.

Glycogen depletion test: 24 h before each test pre- and post-intervention, all participants followed a glycogen-depleting cycling test [31]. Once the test was completed, the participants followed the following distribution of macronutrients in the prescribed diet: 8 g CHO/kg, 2 g PRO/kg, 1 g LIP/kg [32]. Participants were also advised to drink 2.5 L (1 L breakfast, 1 L lunch, 0.5 L dinner) of water to be fully hydrated on the day of the TTE test.

### 2.5. Time-to-Exhaustion Exercise (TTE)

All participants completed the TTE test in the laboratory (in the morning) at the intensity previously found in the MLSS test before and after the training and nutritional 5-week intervention (Figure 1). We used the MLSS intensity due to its relationship with the prediction of performance [18,19], to measure the oxidation of energetic substrates before and after the intervention for the same physiological milestone. All participants reported to the laboratory in the morning in a fed state, having a high-CHO breakfast 2 h before (2 g CHO/kg, 0.4 g PRO/kg, 0.2 g LIP/kg) as the last part of the previously described carbohydrate loading (total: 8 g CHO/kg, 2 g PRO/kg, 1 g LIP/kg) [32]. The participants were given the necessary food (raw, weighed, and separated into different intakes) to carry out their food intake at home before performing the tests. Participants performed the TTE in the laboratory on the ergometer (Wahoo Fitness, model: WF113, Vietnam). The TTE at the previously found MLSS intensity was performed pre- and post-intervention as follows: 5 min warm-up at 100 W followed by 15 min at the individual MLSS intensity and followed by 2 min of recovery at 70% of the MLSS. This protocol was repeated indefinitely until volitional exhaustion [33]. Multiple measurements were collected during this TTE test, including RER, heart rate (HR) (Polar FT1, Helsinki, Finland), body weight before and after exercise test (Tanita BC-730, Tokyo, Japan), cadence, and final lactate levels when participants reached volitional exhaustion. RER and the oxidation of CHO and LIP in g/min were calculated from *V*O_2_ and *VC*O_2_ based on stoichiometric equations [34,35]. Furthermore, data obtained from the gas exchange from the metabolic cart (Ergocard, Medisoft, Sorinnes, Belgium) were obtained after the first 5 min once subjects reached a steady state of RER between 0.9–1 according to the MLSS concept [30]. All subjects were able to maintain said RER during the entire steady-state protocol.

### 2.6. Intervention

#### 2.6.1. Training Protocol

During the 5 weeks (training–diet intervention), both groups completed the same standardized training program that consisted of 5 weeks of cycling and gym sessions with a total volume of 66 h, following criteria of a three-phase model, which was used to create the 5 week training plan; phase 1: 57.5 h, phase 2: 7.5 h, phase 3: 1 h [36]. The participants were provided with their individual training zones for power and heart rate for the bike training, based on the individual’s PPO found in the *GXT* [37]. The participants uploaded the sessions daily to the *Training peaks* platform (285 Century PI, Louisville, CO, USA), which the authors exhaustively reviewed as inclusion criteria.

#### 2.6.2. Nutritional Protocol

All participants followed a controlled diet in which food weighing was mandatory for the five weeks of the training intervention. At the beginning of the study, each participant was given a scale to weigh the food. All participants received personalized feeding plans made depending on the dietary group to which they were assigned and were instructed to follow prescribed menus with similar daily calories but with different amounts of macronutrients/kg of BM and with different timing of intake to ensure a divergent periodization over each day according to the allocated dietary treatment. Common daily-consumption foods (rice, pasta, chicken, oats, etc.) were prescribed, with sufficient variety, to find a high adherence rate. In the PCHO group, the diets were divided into “low”, “medium”, and “high” days following the “fuel for the work required” methodology described by Impey et al. (2018) [17]. On “low” days (bike session Z1-Z2, below VTL1), the participants were deprived of CHO intake from lunch the day before until post-exercise recovery intake (Table 2). The breakfasts were primarily composed of protein and fats, excluding the consumption of CHO. During the training, the food intake consisted only of water with ISO protein. Once the “low” training session was completed, participants then refueled with high quantities of CHO-rich foods and drinks (maltodextrin + fructose) until the next day’s “medium” or “high” training session”. For the “medium” (gym and double-session days) and “high” (bike session Z1-Z5, above VTL1) days, the participants consumed similar amounts of carbohydrates at breakfast, training (gels and isotonic drinks), and recovery meals, while subsequent intakes (lunch, snack, and dinner) were conditioned by the type and intensity of training the following day. The HCHO group consumed foods and/or drinks rich in CHO at all intakes established on the menu. Foods rich in carbohydrates, such as porridge, bread, pasta, etc., were used when preparing the menus. During the training, the participants consumed CHO through gels and isotonic drinks. In recovery, they used quantities of maltodextrin + fructose + ISO protein according to each participant’s menu. Prescribed diets between groups had almost identical daily and weekly energy amounts/kg and were prepared and calibrated with the program Easydiet (Academia Española de Nutrición y Dietética, 2021, Barcelona, Spain). The diet results were interpreted after analyzing records on the MyFitnessPal platform and the Remote Food Photography Method (RFPM) [38]. We found out which participants complied with the established guidelines.

### 2.7. Statistical Analysis

Statistical analyses of data were performed using the Statistical Package for the Social Sciences 21.0 (StatSoft, Tulsa, OK, USA). Data were screened for normality of distribution using a Shapiro–Wilk normality test and all the variables presented a normal distribution (*p* > 0.05). A descriptive analysis of mean and standard deviation was used for all variables for both the PCHO and the HCHO groups. A 2-way repeated-measures ANOVA was performed for all outcome variables to analyze the interaction between groups (PCHO group and HCHO group) and the time of assessment (pre- and post-intervention). When differences were established, we applied a post hoc Bonferroni multiple-comparisons test. Regarding nutritional prescription variables, Student’s *t*-test for independent samples was used to establish the differences between PCHO and HCHO groups. Effect size (ES; Cohen’s d) statistics were used to quantify the magnitude of the difference in pairwise comparisons. The magnitude of the effect size (ES) was interpreted using the Cohen scale as low (~0.2), medium (~0.5), and high (~0.8). Significance for all analyses was set at *p* < 0.05 [39].

## 3. Results

### 3.1. Nutritional Prescription

All participants in the study demonstrated compliance with their assigned dietary treatment and monitoring of their food intake. As intended, there was a different intake pattern according to their group allocation. No differences were detected in total energy prescribed between the PCHO and HCHO groups (*p* > 0.05). The PCHO group consumed, on average, 5.7 CHO/kg of body weight in the first week, which was less than the average of 7.7 CHO/kg consumed by the HCHO group (*p* = 0.026). In the subsequent weeks, there was no difference between the two groups in the CHO intake (*p* > 0.05). When comparing the mean CHO intake for the five weeks, there was a higher consumption in the HCHO group compared to the PCHO group (*p* = 0.001); the CHO intakes for the “low” days for the PCHO group were consumed after the exercise, in their entirety (see example in Table 2). Protein consumption per week and over the 5-week training period was higher in the PCHO group compared to the HCHO group (*p* < 0.05). Lipid intake was higher during each of the first four weeks in the PCHO group compared to the HCHO group (*p* < 0.05), while the difference between groups just failed to reach statistical significance in the fifth week (*p* = 0.053). Overall, when we compared the 5-week means between groups, there was a higher lipid consumption in the PCHO group than in the HCHO group (*p* < 0.001) (Table 3).

### 3.2. Pre–Post-MLSS

In the MLSS, expressed in watts and watts per kg of body mass, a significant time effect (W: F = 11.36, *p* = 0.004; W/kg: F = 37.10, *p* = <0.001) but no significant time × group interaction (W: F = 0.11, *p* = 0.742; W/kg: F = 1.55, *p* = 0.232) or group effect (W: F = 0.39, *p* = 0.542; W/kg: F = 2.05, *p* = 0.172) was found. Both groups improved MLSS tests expressed in both watts (PCHO = 244.1 ± 29.9 to 253.2 ± 28.4 W (CI 95%: from 0.3 to 17.9 W; *p* = 0.043; ES = 0.3); HCHO = 235.8 ± 21.4 to 246.9 ± 16.7 W (CI 95%: from 1.8 to 20.4 W; *p* = 0.022; ES = 0.6) (Figure 2a)) and watts per kg of body mass (PCHO = 3.7 ± 0.5 to 3.9 ± 0.4 W/kg (CI 95%: from 0.2 to 0.4 W/kg; *p* = <0.001; ES = 0.5); HCHO = 3.4 ± 0.3 to 3.7 ± 0.3 W/kg (CI 95%: from 0.1 to 0.3 W/kg; *p* = 0.003; ES = 1.1)).

### 3.3. Pre–Post-Time-To-Exhaustion Exercise (TTE)

However, no significant time × group interaction (F = 0.75, *p* = 0.399), time effect (F = 0.27, *p* = 0.609), or group effect (F = 3.99, *p* = 0.064) was found in the accumulated time as part of the TTE (Figure 2b). Neither group improved their accumulated time during the TTE after the intervention: PCHO = 4745 ± 1534 to 4850 ± 1348 s (CI 95%: from −784.2 to 994.7 s; *p* = 0.804; ES = 0.1); HCHO = 6335 ± 1193 to 5913 ±1869 s (CI 95%: from −1366.0 to 520.8 s; *p* = 0.355; ES = 0.3). No differences between the PCHO and HCHO groups were found in the average cadence during the TTE test, average HR during the TTE test, or final lactate concentration after the TTE test (*p* > 0.05) (Table 4). We found no significant time × group interaction (F = 0.58, *p* = 0.460), time effect (F = 0.03, *p* = 0.878), or group effect (F = 1.28, *p* = 0.275) difference in work in kilojoules (kJ) cost during the performance test (PCHO: 1218.6 ± 471.2 to 1284.6 ± 434.9 KJ; HCHO: 1502.6 ± 384.9 to 1459.2 ± 470.7 KJ; Figure 3).

### 3.4. CHO and LIP Oxidation and Metabolism (RER)

No differences were found in CHO and LIP oxidation during the TTE for both groups (*p* > 0.05). Lastly, there were no changes in RER and VO_2_ at the MLSS for both groups pre- and post-intervention (*p* > 0.05) (Table 4).

## 4. Discussion

The main findings of the present study were as follows: (1) Both groups improved in similar pre–post-MLSS tests expressed in watts and watts/kg. However, the periodized carbohydrate diet did not lead to a greater improvement pre–post-intervention -time-to-exhaustion tests when compared to a traditional high-carbohydrate diet; (2) there were no significant differences in CHO and LIP oxidation and RER during the time-to-exhaustion test at MLSS intensity pre–post-intervention; (3) there was a significant improvement in muscle mass or decrease in fat percentage for both conditions; (4) no significant differences were found between groups during the performance tests in any of the parameters studied (final lactate, mean heart rate, mean cadence, and before and after time-to-exhaustion test and body weight) before or after the intervention.

Based on the current literature, most studies have investigated the relevance of low CHO periodization on well-trained endurance athletes (i.e., cyclists, triathletes, and race walkers with a *V*O_2max_ of ~60 mL/kg/min) for three weeks following different CHO periodizing strategies including “fasted training”, “twice-a-day”, and “sleep-low”. They did not show performance enhancements regarding endurance performance in comparison to high CHO availability after performing a time trial endurance test [8,10,12,16,40]. Even though two studies found beneficial effects of CHO periodization on endurance performance, applying a sleep-low approach during one [9] and three weeks [8], a recent review and meta-analysis, which bring together most of the studies published to date investigating between one and four weeks of periodizing CHO availability in well-trained endurance athletes (≥55–60 *V*O_2max_), revealed that the collective evidence does not support a beneficial or superior approach for enhancing endurance performance compared to a high-CHO diet [41]. The present study extends the period of nutritional diet intervention, applying 5 weeks of intervention because the intervention period (i.e., one to four weeks) is part of the limitations in existing research; more is needed to draw definitive conclusions [9,11]. Another novelty of the present investigation was in performing time-to-exhaustion exercise at steady-state intensity until fatigue to assess the time completed. Detection of MLSS intensity is crucial since a substantial portion of aerobic training in athletes is carried out at MLSS intensities, specifically in well-trained cyclists [19,42]. In the present investigation, both groups improved in the MLSS test pre–post-intervention regardless of the nutritional intervention of the protocol. Furthermore, we did not find any improvement in total time-to-exhaustion exercise in either group (Figure 2a). These results suggest that the MLSS improvement observed in the present study may be due to the training, not the dietary intervention. As described in the study by Mendes et al. (2013), training per se increases the MLSS but does not improve the time to fatigue at MLSS in athletes [43]. This finding suggests that in well-trained athletes, periodized CHO intake does not translate into superior sustained adaptations for enhancing endurance performance compared to a high-CHO diet.

The present study found no differences in substrate utilization and metabolism during the MLSS test (Table 4). Most of the existing studies applied a submaximal test at a certain percentage of the PPO followed by a test until exhaustion [8,9,11,12,14,15], which makes it hard to assess whether the results obtained in the measurements of the use of energy substrates during the submaximal tests were carried out at the same physiological milestone measured with lactate [44]. Therefore, in the present study, we tried to fill this gap by measuring the use of energy substrates for a specific physiological milestone during the TTE test (MLSS intensity, verified with the final lactate of the TTE). Previous researchers obtained similar results to the present study without observing changes in substrate utilization [13,14,16]. In all of them, it was possible to observe no change in the oxidation of substrates after the previous ingestion of a meal rich in CHO (usually carried out before races) before and after main performance tests. For instance, Riis et al. (2019) described that after CHO intake, there is hyperinsulinemia which attenuates the use of fat as fuel in “low” groups [15]. In contrast, other authors reported that carbohydrate and lipid oxidation changes, seeing increased oxidation of lipids at submaximal intensity in groups following a low-carbohydrate, high-fat diet [10,11,12]. Furthermore, studies such as that of Impey et al. (2016) [45] found greater fat oxidation during exercise after carrying out the “sleep-low” strategy. The factors mentioned above, such as previous intake (low in carbohydrates), could condition the oxidation of test substrates, unlike this other study [46]. Lastly, a low-carbohydrate, high-fat diet results in a decrease in the oxidation of CHO during steady-state exercise [11]. This suggests that training with low CHO may be counterproductive for athletes who compete in high-intensity events where CHO oxidation plays a significant role in performance [47]. Finally, despite the different studies that have tried to show that low-carb diets have their place with the premise of increasing fat oxidation during exercise [45] and saving glycogen as a consequence, most of the articles do not endorse it, above all, when a previous intake of CHO (real conditions) is carried out before exercise [48]. Therefore, periodized CHO intake does not show superior metabolism adaptations compared to a high-CHO diet and may also impair training sessions and following training improvements due to lower glucose oxidation and carbohydrate utilization, which may affect the duration and quality of training [2,49,50,51,52].

Notably, we found decreased fat and increased muscle mass in both groups (Table 5). Marquet et al. (2016) [8], matching the caloric intake between groups, obtained significant body fat losses only in the sleep-low (SL) group, which, unlike the current study, was attributed to greater fat oxidation during exercise. In the present study, the general improvement in the anthropometric parameters was as expected since, during the five weeks of intervention, there was a caloric reduction in both groups because the caloric intake was carefully controlled. In contrast to the Marquet study [8], the BM decreased (by ~1 kg) only in the SL group, which was mainly attributed to a 1.1% decrease in fat mass. We found significant weight loss in both groups, which was one of the possible explanations for the performance improvement observed in the SL group of that study. This could suggest that, given the difficulty in controlling the diet, the two groups did not consume the same number of calories because one of the main factors when it comes to improving body composition is the caloric intake in the diet [51].

Lastly, this study faced some limitations. When conducting the research with well-trained participants, obtaining muscle biopsies was denied due to the invasive technique. Therefore, the potential underlying physiological mechanisms were not investigated. Another limiting factor was that the participants had to follow a controlled diet (weighing all foods) and a structured training plan for five weeks, which reduced and limited the number of participants. Furthermore, a periodized CHO diet is not a “natural” nutrition approach to training in competitive athletes due to their ad libitum nutritional characteristics as well as the requirements of both training and competition. Further cellular and molecular studies are warranted to prove any advantages of a periodized CHO intake over a normal or higher CHO diet in well-trained athletes.

## 5. Conclusions

The results of the present study show that periodization of CHO vs. a high-CHO diet during five weeks of supervised exercise training in well-trained athletes does not influence MLSS and does not change substrate oxidation (CHO and LIP) during a time-to-exhaustion test at MLSS intensity. Similarly, it can be concluded that both diets effectively improve anthropometric parameters and exercise performance (watts in MLSS) if caloric intake and training are controlled. Further studies are needed to identify the specific cellular responses to different nutritional interventions and the timing of such interventions deployed to athletes and populations with chronic diseases.

### Practical Applications

HCHO and PCHO diets have a place when planning a diet as long as they are carried out according to prescribed training. Likewise, today, HCHO diets have a greater place in endurance sports where high glycogen concentrations are demanded and in exercises or training where carbohydrate oxidation is essential. It should be considered that low energy availability during prolonged periods may cause adverse effects that will eventually compromise sports performance and health [52]. Lastly, caloric control of the diet is essential as well as providing the athlete with sufficient carbohydrates to withstand the loads and duration of training and competition without harming health and performance.

## Figures and Tables

**Figure 1 nutrients-16-00318-f001:**
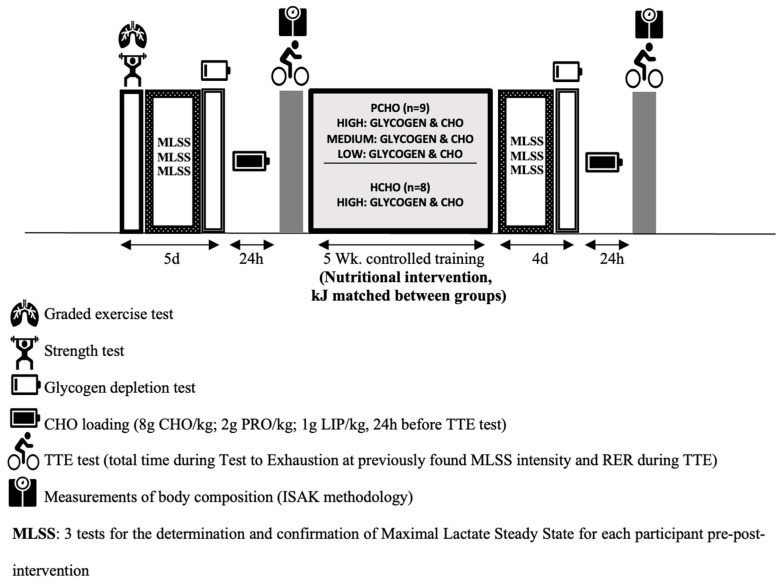
Study design during 5-week nutritional and training intervention. (wk.: week, d: day, MLSS: maximal lactate steady state, RER: respiratory exchange ratio, TTE: time to exhaustion).

**Figure 2 nutrients-16-00318-f002:**
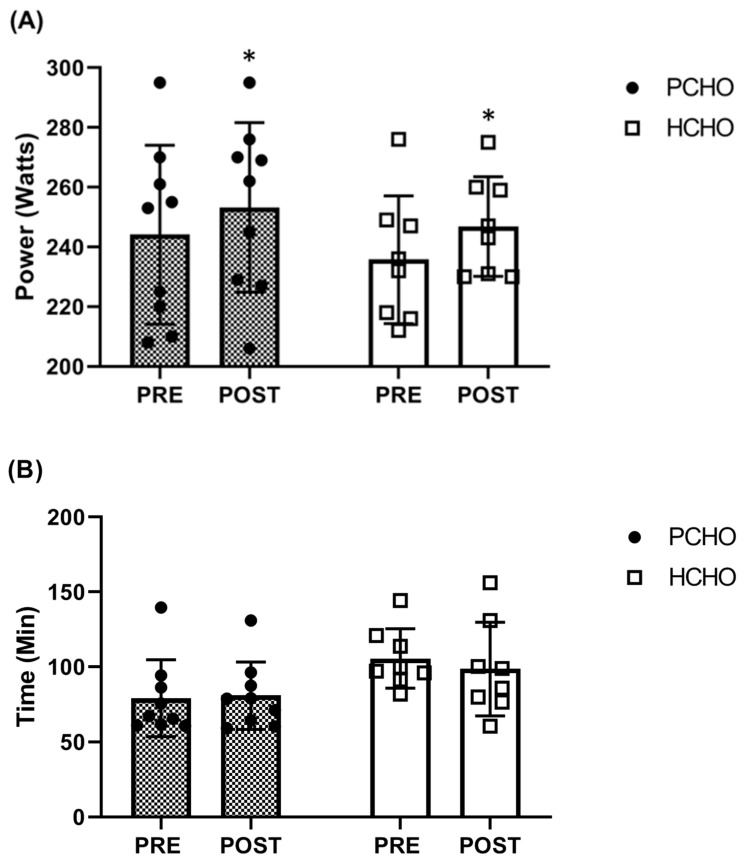
Pre- and post-intervention comparisons of (**A**) the maximal lactate steady state (MLSS) and (**B**) time accumulated during the test to exhaustion at MLSS intensity (TTE) in the high-carbohydrate (HCHO) and periodized carbohydrate (PCHO) groups. Significantly different from pre-treatment: *, *p* < 0.05.

**Figure 3 nutrients-16-00318-f003:**
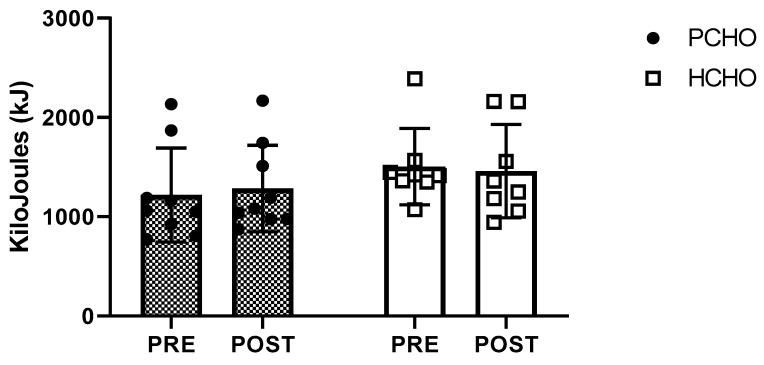
Comparison of pre- and post-intervention of work in kilojoules (kJ) in the high-carbohydrate (HCHO) and periodized carbohydrate (PCHO) groups.

**Table 1 nutrients-16-00318-t001:** Training schedule during 5-week intervention (Zone 1 (Z1) to Zone 7 (Z7) training intensities according to each training power zone and heart rate zone) and type of nutritional plan for the PCHO (low, medium, or high) and HCHO (high) groups. (Wup: warm-up, rec: recovery).

Training	Monday	Tuesday	Wednesday	Thursday	Friday	Saturday	Sunday
*PCHO/HCHO*	Medium/ High	Low/High	High/High	Low/High	Medium/High	High/High	Low/High
*Week 1*	Morning: Gym 1 h 20′ -Wup 15′ + Core 20′ -4 × 4 (55–60 RM)	Morning: Bike 3 h -Ride Z1-Z2	Morning: Bike 2 h 30′ -Wup 30′ -4× (10′Z3 rec 10′Z1-Z2) -40′Z1-Z2	Morning: Bike 2 h 30′ -Ride Z1-Z2	Morning: Gym 1 h 20′ -Wup 15′ + Core 20′ -4 × 4 (55–60 RM)	Morning: Bike 3 h -Ride Z1-Z2	Morning: Bike 2 h 30′ -Ride Z1-Z2
*PCHO/HCHO*	Medium/high	Medium/high	High/high	Low/high	Medium/high	High/high	Low/high
*Week 2*	Rest	Morning: Bike 1 h 30′Z1-Z2 Afternoon: Gym 1 h 20′ -Wup 15′ + Core 20′ -4 × 4 (55–60 RM)	Morning: Bike 3 h 35′ -Wup 30′ -5 × (10′Z3 rec 3′Z1-Z2) -2 h Z1-Z2	Morning: Bike 2 h 30′ -Ride Z1-Z2	Morning: Bike 1 h 30′Z1-Z2 Afternoon: Gym 1 h 20′ -Wup 15′ + Core 20′ -4 × 4 (55–60 RM)	Morning: Bike 3 h -Wup 50′ -5× (10′Z3, rec 6′Z1-Z2) -50′Z1-Z2	Morning: Bike 2 h 30′ -Ride Z1-Z2
*PCHO/HCHO*	Medium/high	Medium/high	High/high	Low/high	Medium/high	High/high	Low/high
*Week 3*	Rest	Morning: Bike 1 h 30′Z1-Z2 Afternoon: Gym 1 h 20′ -Wup 15′ + Core 20′ -4 × 4 (55–60 RM)	Bike 3 h -Wup 50′ -5× (10′Z3 rec 6′Z1-Z2) -50′Z1-Z2	Morning: Bike 2 h 30′ -Ride Z1-Z2	Morning: Bike 1 h 30′Z1-Z2 Afternoon: Gym 1 h 20′ -Wup 15′ + Core 20′ -4 × 4 (55–60 RM)	Morning: Bike 3 h -Wup 60′ -2× Ramp Up (5′Z3, 5′Z4, 5′Z4 5′Z5), rec 20′Z1-Z2 -4× (2′Z5-Z6, rec 8′)	Morning: Bike 2 h 30′ -Ride Z1-Z2
*PCHO/HCHO*	Medium/high	Low/high	Medium/high	Low/high	Medium/high	High/high	Low/high
*Week 4*	Rest	Morning: Bike 2 h 30′ -Ride Z1-Z2	Morning: Bike 1 h 30′Z1-Z2 Afternoon: Gym 1 h 20′ -Wup 15′ + Core 20′ -4 × 4 (55–60 RM)	Morning: Bike 2 h 30′ -Ride Z1-Z2	Morning: Bike 1 h 30′Z1-Z2 Afternoon: Gym 1 h 20′ -Wup 15′ + Core 20′ -4 × 4 (55–60 RM)	Morning: Bike 3 h -Wup 60′ -2× Ramp Up (5′Z3, 5′Z4, 5′Z4 5′Z5), rec 20′Z1-Z2 -4× (2′Z5-Z6, rec 8′)	Morning: Bike 2 h 30′ -Ride Z1-Z2
*PCHO/HCHO*	Medium/high	Medium/high	Low/high	High/high	Low/high	Medium/high	Low/high
*Week 5*	Rest	Morning: Bike 1 h 30′Z1-Z2 Afternoon: Gym 1 h 20′ -Wup 15′ + Core 20′ -4 × 4 (55–60 RM)	Morning: Bike 2 h 30′ -Ride Z1-Z2	Morning: Bike 2 h 30′ -Wup 30′Z1-Z2 -60′Z3 -30′Z1-Z2	Morning: Bike 3 h -Ride Z1-Z2	Rest	Morning: Bike 2 h 30′ -Ride Z1-Z2

**Table 2 nutrients-16-00318-t002:** Example of the first week of intervention for the PCHO and HCHO groups. Distribution of macronutrients for the different meal intakes (g/kg of BM) except for the training expressed as total macronutrient intake. (classif.: classification, macro.: macronutrients, wk: week, recov.: recovery meal, BM: body mass, MED.: medium, CHO: carbohydrate, PRO: protein, LIP: lipid, kJ: kilojoule, kg: kilogram).

WK 1	Day 1		Day 2		Day 3		Day 4		Day 5		Day 6		Day 7	
	**PCHO**	**HCHO**	**PCHO**	**HCHO**	**PCHO**	**HCHO**	**PCHO**	**HCHO**	**PCHO**	**HCHO**	**PCHO**	**HCHO**	**PCHO**	**HCHO**
Classif.	MED		LOW		HIGH		LOW		MED.		HIGH		LOW	
Macro.	CHO/PRO/LIP	CHO/PRO/LIP	CHO/PRO/LIP	CHO/PRO/LIP	CHO/PRO/LIP	CHO/PRO/LIP	CHO/PRO/LIP	CHO/PRO/LIP	CHO/PRO/LIP	CHO/PRO/LIP	CHO/PRO/LIP	CHO/PRO/LIP	CHO/PRO/LIP	CHO/PRO/LIP
Breakf.	2.4/0.5/0.2	1.1/0.5/0.1	0.0/0.4/0.2	2.0/0.2/0.1	3.0/0.5/0.3	2.0/0.4/0.3	0.0/0.4/0.2	2.0/0.2/0.1	1.1/0.5/0.1	1.1/0.5/0.1	2.8/0.6/0.4	2.8/0.3/0.2	0.0/0.4/0.2	2.0/0.2/0.1
Train.	Gym 1 h 20′ 30 g CHO (23 g/h)	Gym 1 h 20′ 30 g CHO (23 g/h)	Bike 3 h Z1-Z2 45 g PRO (15 g/h)	Bike 3 h Z1-Z2 90 g CHO (30 g/h)	Bike 2 h Z2-Z4 100 g CHO (50 g/h)	Bike 2 h Z2-Z4 100 g CHO (50 g/h)	Bike 2 h 30′ Z1-Z2 37.5 g PRO (15 g/h)	Bike 2 h 30′ Z1-Z2 75 g CHO (30 g/h)	Gym 1 h 20′ 30 g CHO (23 g/h)	Gym 1 h 20′ 30 g CHO (23 g/h)	Bike 3 h Z2 150 g CHO (50 g/h)	Bike 3 h Z2 150 g CHO (50 g/h)	Bike 2 h 30′ Z1-Z2 37.5 g PRO (15 g/h)	Bike 2 h 30′ Z1-Z2 75 g CHO (30 g/h)
Recov.	1.2/0.5/0.0	1.0/0.5/0.0	1.2/0.3/0.0	1.0/0.3/0.0	1.2/0.5/0.0	1.0/0.5/0.0	1.2/0.3/0.0	1.0/0.3/0.0	1.0/0.5/0.0	1.0/0.5/0.0	1.2/0.5/0.0	1.0/0.3/0.0	1.2/0.4/0.0	1.0/0.3/0.0
Lunch	0.3/0.5/0.4	1.0/0.5/0.2	1.3/0.6/0.2	1.0/0.4/0.2	0.3/0.6/0.4	1.0/0.5/0.3	1.2/0.5/0.3	1.0/0.4/0.2	1.0/0.6/0.2	1.0/0.6/0.2	0.5/0.6/0.6	2.0/0.4/0.3	1.0/0.6/0.3	1.0/0.4/0.2
Snack	0.3/0.5/0.2	1.0/0.5/0.1	1.1/0.2/0.1	0.6/0.2/0.1	0.0/0.4/0.3	1.0/0.3/0.2	1.0/0.2/0.1	0.4/0.3/0.0	0.9/0.2/0.2	0.9/0.2/0.2	0.3/0.6/0.5	1.7/0.3/0.0	0.8/0.3/0.2	0.4/0.3/0.0
Dinner	0.3/0.5/0.4	2.0/0.5/0.2	2.0/0.5/0.3	1.4/0.4/0.1	0.3/0.7/0.5	1.2/0.5/0.3	1.7/0.5/0.2	1.0/0.4/0.2	1.2/0.6/0.3	1.2/0.6/0.3	0.5/0.7/0.6	2.0/0.4/0.3	1.0/0.6/0.4	1.0/0.4/0.2
Daily CHO/PRO/LIP /kg BM	5.0/2.5/1.2	6.6/2.5/0.5	5.6/2.8/0.8	7.6/1.5/0.5	6.6/2.7/1.5	8.0/2.2/1.1	5.1/2.6/0.8	6.8/1.6/0.5	5.7/2.5/0.9	5.7/2.5/0.9	8.0/3.0/2.1	12.2/1.7/0.8	4.0/3.0/1.1	6.8/1.6/0.5
Daily kJ/kg	170.7	171.1	170.7	171.1	212.1	212.1	159	159	171.1	171.1	263.2	262.8	158.6	159.4

**Table 3 nutrients-16-00318-t003:** Macronutrient and kilojoule average (kJ) intake per kilogram (kg) of the intervention per week and total for the high-carbohydrate (HCHO) and periodized carbohydrate (PCHO) groups.

	KJ (kJ/kg)	CHO (g/kg)	PRO (g/kg)	LIP (g/kg)
**Week 1**				
PCHO	186.6 ± 38.1	5.7 ± 1.3 *	2.7 ± 0.2 *	1.2 ± 0.5 *
HCHO	187.0 ± 37.7	7.7 ± 2.1 *	1.9 ± 0.4 *	0.7 ± 0.3 *
**Week 2**				
PCHO	202.1 ± 53.6	6.3 ± 2.4	2.7 ± 0.3 *	1.4 ± 0.4 *
HCHO	202.1 ± 52.7	8.3 ± 3.1	1.9 ± 0.4 *	0.8 ± 0.3 *
**Week 3**				
PCHO	202.1 ± 53.6	6.3 ± 2.4	2.7 ± 0.3 *	1.4 ± 0.4 *
HCHO	202.1 ± 52.6	8.3 ± 3.1	1.9 ± 0.4 *	0.8 ± 0.3 *
**Week 4**				
PCHO	179.1 ± 39.3	5.2 ± 1.7	2.7 ± 0.3 *	1.2 ± 0.5 *
HCHO	179.5 ± 38.9	7.2 ± 2.1	1.9 ± 0.4 *	0.7 ± 0.3 *
**Week 5**				
PCHO	169.0 ± 30.1	4.7 ± 1.3	2.6 ± 0.3 *	1.2 ± 0.5
HCHO	169.5 ± 29.7	6.6 ± 1.2	1.8 ± 0.3 *	0.8 ± 0.3
**Week 1 to 5 (total)**				
PCHO	187.9 ± 43.1	5.7 ± 1.9 *	2.7 ± 0.3 *	1.3 ± 0.4 *
HCHO	187.9 ± 42.7	7.6 ± 2.4 *	1.8 ± 0.3 *	0.8 ± 0.3 *

Significantly different between groups: *, *p < 0.05.*

**Table 4 nutrients-16-00318-t004:** Comparison of pre- and post-intervention of respiratory exchange ratio (RER), carbohydrate (CHO) and lipid (LIP) oxidation, oxygen volume (*V*O_2_), heart rate (HR), average cadence, and final lactate during the test to exhaustion and body mass loss after the test to exhaustion in the high-carbohydrate (HCHO) and periodized carbohydrate (PCHO) groups.

	PRE	POST		*F*	*p*	*ƞ* *p* ^2^
**CHO oxidation (g/min)**						
PCHO	3.92 ± 0.75	3.78 ± 0.94	Group	0.98	0.339	0.06
HCHO	4.00 ± 0.67	4.38 ± 0.81	Time	0.37	0.553	0.02
			Group × Time	1.87	0.192	0.11
**LIP oxidation (g/min)**						
PCHO	0.34 ± 0.21	0.37 ± 0.19	Group	3.27	0.091	0.18
HCHO	0.23 ± 0.22	0.16 ± 0.23	Time	0.16	0.693	0.01
			Group × Time	0.72	0.410	0.05
**RER (*V*CO_2_/*V*O_2_)**						
PCHO	0.94 ± 0.03	0.94 ± 0.04	Group	4.54	0.050	0.23
HCHO	0.96 ± 0.04	0.99 ± 0.05	Time	0.54	0.475	0.04
			Group × Time	2.38	0.144	0.14
***V*O_2_ (ml/kg/min)**						
PCHO	3384 ± 395	3573 ± 453	Group	0.81	0.383	0.05
HCHO	3421 ± 365	3421 ± 359	Time	0.01	0.923	0.01
			Group × Time	0.01	0.908	0.01
**Cadence (rpm)**						
PCHO	84.89 ± 6.21	84.44 ± 6.19	Group	3.84	0.069	0.20
HCHO	79.25 ± 9.60	76.00 ± 10.62	Time	1.06	0.320	0.07
			Group × Time	0.61	0.447	0.04
**HR (bpm)**						
PCHO	151.22 ± 8.23	150.44 ± 9.93	Group	0.10	0.755	0.01
HCHO	154.25 ± 3.49	145.38 ± 3.49	Time	4.86	0.050	0.25
			Group × Time	3.42	0.084	0.19
**Lactate (mmol/L)**						
PCHO	3.86 ± 1.47	3.46 ± 0.70	Group	1.10	0.312	0.07
HCHO	3.19 ± 1.12	3.39 ± 1.03	Time	0.06	0.812	0.01
			Group × Time	0.53	0.480	0.03
**Body mass loss (%)**						
PCHO	2.42 ± 0.53	2.51 ± 0.64	Group	2.43	0.140	0.14
HCHO	2.24 ± 0.37	1.92 ± 0.86	Time	0.41	0.532	0.03
			Group × Time	1.31	0.271	0.08

*ƞp*^2^, partial eta squared; Group, main effect of group in the ANOVA results; Time, main effect of time in the ANOVA results; Group × Time, main effect of group × time in the ANOVA results.

**Table 5 nutrients-16-00318-t005:** Comparison of pre- and post-intervention body mass, % of muscle mass, % of body fat, and the sum of 6 skinfolds in the high-carbohydrate (HCHO) and periodized carbohydrate (PCHO) groups.

	PRE	POST		*F*	*p*	*ƞ* *p* ^2^
**Body Mass (kg)**						
PCHO	67.08 ± 6.07	65.77 ± 5.11 *	Group	1.11	0.309	0.07
HCHO	70.55 ± 5.39	67.71 ± 4.73 *	Time	25.58	<0.001	0.63
			Group × Time	3.46	0.082	0.19
**Muscle Mass (%)**						
PCHO	50.26 ± 1.23	50.77 ± 1.32 *	Group	0.12	0.731	0.01
HCHO	49.98 ± 1.58	50.60 ± 1.21 *	Time	21.91	<0.001	0.59
			Group × Time	0.22	0.644	0.02
**Body Fat (%)**						
PCHO	8.32 ± 1.15	7.49 ± 1.33 *	Group	0.16	0.699	0.01
HCHO	8.79 ± 1.23	7.46 ± 1.0 *	Time	70.26	<0.001	0.82
			Group × Time	3.84	0.069	0.20
**Sum 6 skinfolds (mm)**						
PCHO	48.16 ± 11.87	39.70 ± 13.65 *	Group	0.16	0.697	0.01
HCHO	53.06 ± 12.68	39.39 ± 10.27 *	Time	71.63	<0.001	0.83
			Group × Time	3.98	0.064	0.21

*ƞp*^2^, partial eta squared; Group, main effect of group in the ANOVA results; Time, main effect of time in the ANOVA results; Group × Time, main effect of group × time in the ANOVA results. *, significantly different from pre-treatment with *p* < 0.05.

## Data Availability

Data are contained within the article.

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
