# Peer review of "A Five-Week Periodized Carbohydrate Diet Does Not Improve Maximal Lactate Steady-State Exercise Capacity and Substrate Oxidation in Well-Trained Cyclists compared to a High-Carbohydrate Diet"

_nutrients, 2024, doi:10.3390/nu16020318_

Round 1

Reviewer 1 Report

Comments and Suggestions for Authors

Dear Research Team,

I read with great interest your recent project on CHO periodization. It is a well done study in a cohort that I consider difficult to recruit for a study such as this. While the study definitely has merit in today's individualized nutrition paradigm, there is work that is required to enhance its readability.

Introduction:

There are instances where variables are just mentioned without any justification for their inclusion in this study. Substrate utilization, body composition are the primary two identified in the research question, but almost no background on why these are of interest nor why they should be. Please elaborate in the introduction for the rationale regarding these two variables.

Methods:

Explain how they were randomized to groups.

Why 13 low sessions? Seems this should have been more provided the justification for the study. This should be explained better in the methods.

Lines 83-84: Please clarify these sentences. It is unclear what exactly is trying to be stated here.

VO2max: I see criteria were provided, but how many actually achieved this? Please include this. Even in elite cyclists, I have not found all of them to achieve these criteria before volitional exhaustion.

Lines 122: Should not or did not? Several instances throughout the paper where the authors are writing in future tense rather than past tense. Please check for consistency.

Line 129 and Line 140: Again consistency throughout. Some instances it is FAT and others are LIP (also see your diet Table).

Line 130: should be 0.5 L not 0,5L

Line 136: remove above all and elsewhere it occurs

Line 153: In most studies, the last 5 min of expired air are included so that there is no carryover from earlier stages. Why was the first 5 min included here? If there is no strong justification, then I consider strongly the research team go back and analyze the last 5 min of expired air instead of the first 5 min. This project may likely be reporting a Type 2 error with their CHO/FAT oxi data. When reviewing the data, see group effects for RER (0.05) and LIP oxi (0.09).

Line 185: doble should read double

Figure 2: Vo2 should read VO2

Figures and Tables look good and are presented well.

Need citation with Cohens D scale.

Discussion

It is interesting that muscle mass and BF% are discussed and yet, neither are reported in the results. This must be included in the results to then discuss later.

Line 329: Athletes should reflect the cohort, which were cyclists (although well trained for sure).

Line 347: No need to introduce new abbreviations here (ie., LCHF). Simply spell it out since its only used twice.

While direct measures of metabolism are missing and identified in the limitations section, there are other limitations that should be addressed. The nutrition intervention was well controlled, but is also lacking ecological validity. How does this actually reflect this populations eating habits, which are ad lib in nature. How could future research build upon this limitation and include it.

Line 385: I would suggest stating anthropometric measures without impairing performance...

Overall, a well executed study and has some promise for nutrition research.

Comments on the Quality of English Language

While the actual use of the English language is sufficient, there are multiple grammatical errors that occur throughout this manuscript. Several appearances of abbreviations without first being spelled (examples: Line 41, 84, 110, 114, etc). There are inconsistencies that occur throughout between used abbreviations or not used (CHO vs. carbohydrate; heart rate vs. HR; periodized CHO vs. PCHO) etc. This needs quite a bit of work and review before being acceptable from a writing perspective.

Author Response

The authors would like to thank the Editors and the Reviewers for their helpful comments/suggestions and assistance in improving the quality of the manuscript. The Reviewers' remarks have been addressed in this response letter. Changes within the document have been highlighted (in red; deletions have not been tracked). We genuinely believe our paper is improved in light of the suggested changes.

Reviewer 1, comments:

I read with great interest your recent project on CHO periodization. It is a well done study in a cohort that I consider difficult to recruit for a study such as this. While the study definitely has merit in today's individualized nutrition paradigm, there is work that is required to enhance its readability.

We thank the Reviewer for the interesting and valuable considerations and suggestions. We have improved the rationality and the potential of the manuscript derived from your analysis.

Introduction:

There are instances where variables are just mentioned without any justification for their inclusion in this study. Substrate utilization, body composition are the primary two identified in the research question, but almost no background on why these are of interest nor why they should be. Please elaborate in the introduction for the rationale regarding these two variables.

Thanks for your consideration. We agree with the Reviewer’s recommendation.  We have now included this paragraph in the introduction section: “Regarding substrate utilization and body composition, one of the most interesting and discussed outcomes when applied to CHO-restricted diets is increasing fat as fuel utilization to enhance endurance sports performance [11]. However, significant fat oxidation capacity improvements have been shown to be insufficient to observe improved performance in elite athletes compared with a high CHO condition [13]. Further, low-carbohydrate diets are often used as weight-loss strategies by exercising individuals and athletes [8]. Several researchers reported a decrease in fat-free mass after low-carbohydrate diets for extended periods (> six months) [20]. However, when both interventions were performed in a short-term period (e.g., three weeks) with analogous daily calorie intake, markers of body composition were altered to a similar extent [21]”

Methods

Explain how they were randomized to groups.

We thank the Reviewer for the comment. Firstly, all participants were randomly assigned using the tool https://www.randomizer.org. Once cyclists were assigned to the HCHO and PCHO groups, we checked that there were no significant differences between VO2 peak, body weight, and PPO, see Table 5.

Why 13 low sessions? Seems this should have been more provided the justification for the study. This should be explained better in the methods.

We thank the Reviewer for the recommendation. Now it reads:  The participants performed between two and three low intensity sessions per week based on other studies which found significant changes in performance through this training protocol [8,17]. Furthermore, nutritional plans were prepared according to the exercise intensity schedule that was prescribed. Low intensity sessions were performed at the training intensity described as Zone 1 (Z1) -Zone 2 (Z2), which was determined by the exercise intensity below the first lactate/ventilatory threshold, VTL1. Subjects did not have two low intensity sessions consecutively (Table 1).

Lines 83-84: Please clarify these sentences. It is unclear what exactly is trying to be stated here.

We thank the Reviewer for the suggestion. As mentioned above, we have included a new paragraph in the methods section to address this comment.

VO2max: I see criteria were provided, but how many actually achieved this? Please include this. Even in elite cyclists, I have not found all of them to achieve these criteria before volitional exhaustion.

We thank the Reviewer for the recommendation. The reviewer was right. For elite cyclists, it isn't easy to achieve those criteria. We have redefined the test as Graded Exercise Test (GXT) and changed the VO2 Max for the VO2 Peak, which is what cyclists reached. That being said, we believe it would be more appropriate to name this parameter VO2peak instead. This has been corrected in the manuscript accordingly.

Lines 122: Should not or did not? Several instances throughout the paper where the authors are writing in future tense rather than past tense. Please check for consistency.

We thank the Reviewer for the correction. It has been changed.

Line 129 and Line 140: Again consistency throughout. Some instances it is FAT and others are LIP (also see your diet Table).

We thank the Reviewer for the correction. We have changed FAT to LIP

Line 130: should be 0.5 L not 0,5L

We thank the Reviewer for the correction. We have changed 0,5L to 0.5 L.

Line 136: remove above all and elsewhere it occurs

We thank the Reviewer for the correction which we have made.

Line 153: In most studies, the last 5 min of expired air are included so that there is no carryover from earlier stages. Why was the first 5 min included here? If there is no strong justification, then I consider strongly the research team go back and analyze the last 5 min of expired air instead of the first 5 min. This project may likely be reporting a Type 2 error with their CHO/FAT oxi data. When reviewing the data, see group effects for RER (0.05) and LIP oxi (0.09).

We thank the Reviewer for the correction. We made a mistake in the description of the protocol methodology, which we have corrected. Instead, we meant that gas exchange was collected after the first 5 minutes, once subjects reached a steady state of RER between 0.9-1 according to the MLSS concept [30]. All subjects were able to maintain that RER during the entire steady-state protocol, as seen in Table 4. We hope the Reviewer is satisfied with our response.

Line 185: doble should read double

We thank the Reviewer for the recommendation, which has been corrected.

Figure 2: Vo2 should read VO2

We thank the Reviewer for the recommendation, which has been corrected.

Need citation with Cohens D scale.

We thank the Reviewer for the recommendation, which has been corrected.

Discussion

It is interesting that muscle mass and BF% are discussed and yet, neither are reported in the results. This must be included in the results to then discuss later.

Thanks for your consideration. These outcomes (muscle mass and %BF) are included in Table 5. We have not included them in the results sections to avoid duplicating information. We hope the Reviewer is pleased with our response.

Line 329: Athletes should reflect the cohort, which were cyclists (although well trained for sure).

We thank the Reviewer for the suggestion. Now it reads; “Based on the current literature, most studies have investigated the relevance of low CHO periodization on well-trained endurance athletes (i.e., cyclists, triathletes and race walkerswith a VO2max of ~ 60ml/kg/min) during three weeks following different CHO periodizing strategies including "fasted training," "twice-a-day," and "sleep low”. They did not show performance enhancements on endurance performance in comparison to high CHO availability after performing a time trial endurance test [8,10,12,16,41]”.

Line 347: No need to introduce new abbreviations here (ie., LCHF). Simply spell it out since its only used twice.

We thank the Reviewer for the correction which we have corrected.

While direct measures of metabolism are missing and identified in the limitations section, there are other limitations that should be addressed. The nutrition intervention was well controlled, but is also lacking ecological validity. How does this actually reflect this populations eating habits, which are ad lib in nature. How could future research build upon this limitation and include it.

Thank you for the comment. We agree with the Reviewer and we have added the following sentence:

“Furthermore, a periodized CHO diet is not a "natural" nutrition approach to training in competitive athletes due to their ad libitum nutritional characteristics as well as the requirements of both training and competition. Further cellular and molecular studies are warranted to prove any advantages of a periodized CHO intake over a normal or higher CHO diet in well-trained athletes”. We hope that the Reviewer is pleased with this statement.

Line 385: I would suggest stating anthropometric measures without impairing performance.

We thank the Reviewer for the suggestion and agree. We have changed this paragraph in order to improve its understanding. Now it reads: “Similarly, it can be concluded that both diets effectively improve anthropometric parameters and exercise performance (Watts in MLSS) if caloric intake and training are controlled.”

Reviewer 2 Report

Comments and Suggestions for Authors

Thank you for submitting the manuscript “A 5-week carbohydrate periodized diet does not improve maximal lactate steady state exercise capacity and substrate oxidation in well-trained cyclists compared to a high carbohydrate 4 diet” to Nutrients. The manuscript appears to have been well conducted and even used methods such as lactate measurement to increase the reliability of the study. Furthermore, the manuscript is well written, the methodology is repeatable and the results were well explored. In fact, the number of research participants was small although it was designed with a much more homogeneous group. I have some suggestions:

- It is not necessary to separate the abstract into topics. This happens naturally throughout the text. Also, it is necessary to include a justification of why the study is important.

- Figures and tables should be as close as possible to where they are first cited in the text. I suggest that authors try to organize these elements in a better way in the text to make it easier to read.

- Table 2: on day 5, carbohydrate intake is the same for both groups, is this correct? As this table is just an example, it could be included in the manuscript's supplementary files.

- Table 3: in week 3 the caloric intake seems to have a difference, is this value correct?

- It would be interesting if the manuscript had in the title or abstract how much was the average difference in percentage of the low and high diet to present this information to the reader in advance.

- Line#221: do you mean in relation to the total energy consumed or prescribed? Consider checking this entire sub-item as you are unsure whether the content was consumed or prescribed.

Author Response

The authors would like to thank the Editors and the Reviewers for their helpful comments/suggestions and for their assistance in improving the quality of the manuscript. The Reviewers' remarks have been addressed in this response letter. Changes within the document have been highlighted (in red; deletions have not been tracked). We genuinely believe that our paper is improved in light of the suggested changes.

It is not necessary to separate the abstract into topics. This happens naturally throughout the text. Also, it is necessary to include a justification of why the study is important.

We thank the Reviewer for the suggestion and agree. We have removed the sections in the abstract and included a justification. Now it reads: “There is a growing interest in studies involving carbohydrate (CHO) manipulation and subsequent adaptations to endurance training”.

Figures and tables should be as close as possible to where they are first cited in the text. I suggest that authors try to organize these elements in a better way in the text to make it easier to read.

We thank the reviewer for the suggestion and agree. Unfortunately, we cannot make any changes because the format required by the journal populates it automatically based on the information provided. If the Reviewer has a specific suggested approach, we would be happy to include it.

Table 2: on day 5, carbohydrate intake is the same for both groups, is this correct? As this table is just an example, it could be included in the manuscript's supplementary files.

We thank the Reviewer for the comment. Yes, that value is correct.

Table 3: in week 3 the caloric intake seems to have a difference, is this value correct?

We thank the Reviewer for the comment. In fact, that value is not correct. We have changed to the correct one, “202.1 ± 52.6”. Thank you again for the correction.

It would be interesting if the manuscript had in the title or abstract how much was the average difference in percentage of the low and high diet to present this information to the reader in advance.

We thank the Reviewer for the suggestion. We suggest not changing the article's title or abstract because when designing the nutritional plan, we have considered the carbohydrate intake, which varies for different intakes, following a pattern of low, medium, or high carbohydrates (obtaining different percentages). The nutritional periodization guidelines (PCHO) have been designed following the guidelines provided by other authors.  Hence, the energy and macronutrient results vary as expected while maintaining a similar caloric intake. The essential element in our study herein was how the distribution of macronutrients was determined for the different prescribed CHO intakes. We hope that the Reviewer is satisfied with our answer.

Line#221: do you mean in relation to the total energy consumed or prescribed? Consider checking this entire sub-item as you are unsure whether the content was consumed or prescribed.

We thank the Reviewer for the question. In our study, we referred to the total energy prescribed in the diet. Subsequently, as we reflect on the study, all intakes were corroborated through the MyFitnessPal platform and the Remote Food Photography Method (RFPM) [39] of each participant.

On behalf of this study's authors, thank you for your constructive comments.